# Dietary Supplementation with Complex Enzymes and Tea Residue Improved the Production Efficiency of Xiangling Pigs

**DOI:** 10.3390/ani15091229

**Published:** 2025-04-27

**Authors:** Runhua Yang, Yulian Li, Zhenyu Lei, Maisheng Wu, Hong Tan, Fang Liu, Yanmei Gong, Weijian Zhong, Jiayan He, Shujuan Zeng, Zhiyong Fan, Shusong Wu

**Affiliations:** 1College of Animal Science and Technology, Hunan Agricultural University, Changsha 410128, China; 2Xiangtan Livestock Breeding Station, Xiangtan 411104, China; 3College of Animal Science and Technology, Sichuan Agricultural University, Chengdu 611137, China; 4Xiangtan LiHua Animal Husbandry Co., Ltd., Xiangtan 411232, China

**Keywords:** tea residue, complex enzymes, caffeine, performance, serum biochemistry, meat quality, intestinal microbiota

## Abstract

This study explored how tea residue combined with complex enzymes impacts Xiangling fattening pigs. 120 healthy pigs, averaging 47.93 kg, were divided into five groups: a control group on a basal diet and four experimental groups with 5.8% fermented tea residue and different levels of complex enzymes (0, 200, 400, and 800 mg/kg). After 83 days, results showed that the combination didn’t affect growth performance. However, tea residue reduced certain liver enzyme activities in the serum, while complex enzymes decreased cholesterol levels and another liver enzyme activity. For meat quality, tea residue altered pH and color values, and complex enzymes affected color and shear force. Amino acid content increased with both treatments, and the 800 mg/kg enzyme group had changes in fatty acid composition. Microbial analysis indicated that tea residue increased the abundance of some bacteria, and enzyme supplementation enriched specific microbial families. Overall, the combination of tea residue and complex enzymes improved pork quality, enhanced metabolic health, and adjusted gut microbiota, with the 400 mg/kg enzyme dose showing the best results. This offers a new feeding approach to boost pork quality without sacrificing growth.

## 1. Introduction

As a vital component of China’s modern agriculture, the healthy development of anaimal husbandry holds significant strategic importance for the supply of national animal products, rural development, the enhancement of farmers’ incomes, and national food security [1]. Pig farming, a cornerstone of the country’s animal husbandry sector, plays a crucial role in the agricultural modernization system. However, the acute shortage of protein resources for feed and the low utilization of bulk non-grain protein resources have hindered industry modernization [2]. In 2023, Hunan Province ranked fourth nationally in the comprehensive output value of the tea industry chain, boasting a tea plantation area of 3.52 million mu and an annual output of 338,000 tons [3]. Tea residue is considered to be an environmentally sustainable and economically viable agricultural by-product derived mainly from tea processing and is nutrient-rich, containing protein, carbohydrates, and fats. Additionally, tea residue also contains various functional components, like polyphenols, flavonoids, and so on. Therefore, tea residue is regarded as a promising candidate for feed material due to its nutritional value and low price, as well as good functional properties [4]. However, owing to the dearth of rational and effective means of disposal, tea residue is often directly discarded or burned, leading to serious environmental pollution and the waste of resources. Existing research indicates that these bioactive compounds possess antioxidant and antibacterial properties, positively influencing intestinal health in animals [5].

While woody plant leaves offer substantial nutritional benefits, their utilization in animal feed is restricted by the abundance of fiber and anti-nutritional components, which adversely affect feed palatability and digestive efficiency. In general, the content of anti-nutritional factors is an important indicator to determine whether the leaves are suitable for animal feed [6]. Complex enzymes, commonly utilized as feed additives, enhance the digestibility and utilization efficiency of feed by breaking down non-starch polysaccharides and other indigestible components [7].

The Xiangling pig, an exemplary local breed in China, is renowned for its tolerance to rough feeding, high-quality meat, and distinctive flavor [8]. However, its slow growth rate and the scarcity of conventional protein sources have constrained its development within the industry. Given the recent emphasis placed on local pig germplasm resources by national policies, it is essential to explore unconventional feed ingredients and nutritional strategies to enhance the growth performance and meat quality of Xiangling fattening pigs. However, until now, little was known regarding the efficacy of tea residue in the pig industry. Considering the benefits, we hypothesized that the partial replacement of corn with tea residue in the diet might have a positive impact on the growth performance, meat quality, fatty acid profile, and antioxidant capacity of Xiangling pigs. To test the hypothesis, this study was conducted to investigate the effects of tea residue on the growth performance, meat quality, muscle fatty acid profile, antioxidant capacity, and intestinal microbiota of Xiangling pigs. Our study aimed to provide a theoretical foundation for the potential implementation of tea residue and complex enzymes in Xiangling pig production and, more broadly, in the pig industry.

## 2. Materials and Methods

### 2.1. Experimental Materials

In September and November 2023, fresh tea residue required for the test was collected in the pilot workshop of Hunan Agricultural University. Tea residue is the residue after the active substances are extracted from black tea (seasonal tea, mechanically harvested tea, containing a small amount of tea stems), which is dried and crushed for feed preparation. The nutrient components and anti-nutritional factors in the tea residue were determined, as shown in Table 1. Complex enzymes (cellulase, 3000 U/g; xylanase, 10,000 U/g; alkaline protease, 10,000 U/g; tannase, 100 U/g) were purchased from Xijie Youtel Biotechnology Co., Ltd. (Zoucheng, China).

### 2.2. Experimental Design

A total of 120 castrated males Xiangling pigs, averaging 47.93 kg (±15.28 kg), were randomly assigned to five treatment groups, each comprising four replicates of six pigs. The average weight per replicate was consistent at 47.93 kg (±0.08 kg). The control group (CON) received a basal diet; the experimental groups received a diet containing 5.8% fermented tea residue as an alternative energy and protein source (dry matter basis) and mixed additional 0 (CZ), 200 (M200), 400 (M400), and 800 (M800) complex enzymes. Both the basal and tea residue diets were formulated in accordance with the “Nutritional Requirements of Chinese Pigs” (GB/T 39235-2020) (China, 2020), meeting the nutritional needs of meat–fat pigs weighing 60–90 kg. The composition and nutritional levels of the experimental diets are detailed in Table 2. Both diets maintained identical energy-to-nitrogen ratios and provided equivalent levels of lysine, methionine, threonine, and tryptophan (all values correspond to calculated total amino acids), while other amino acids met the recommended nutritional standards. The experimental period lasted for 83 days.

### 2.3. Feeding and Management

The experiment was conducted at the Shaziling Pig Research Institute in Xiangtan County, Hunan Province, from October to December 2023. The feeding, immunization, and management procedures for the experimental animals adhered to the standardized protocols of the pig farm. All pigs were housed in fattening pig barns, with six pigs per pen, across a total of 20 pens. They were fed twice daily at 08:00 and 16:00, ensuring a slight surplus of feed remained in the troughs after feeding. The pigs had free access to water. The pigsty was cleaned daily, with feces removed to maintain a dry and hygienic environment.

### 2.4. Determination Indicators and Methods

#### 2.4.1. Growth Performance

One day before the start of the formal experiment and on the final day, the fasting body weight of each experimental pig was recorded to determine the initial body weight (IBW) and final body weight (FBW). The remaining feed for each group was weighed at both the beginning and end of the experimental period to calculate the average daily feed intake (ADFI).

#### 2.4.2. Serum Biochemical Indices

Five milliliters of blood was collected from the jugular vein of the test pigs using a disposable blood collection tube at slaughter. The serum was separated after 1 h of refinement at 4 °C and centrifuged at 3000 rpm for 10 min, then stored at −20 °C for testing. Serum triglyceride (TG), total cholesterol (TC), high-density lipoprotein cholesterol (HDL-C), low-density lipoprotein cholesterol (LDL-C), aspartate aminotransferase (AST), alanine aminotransferase (ALT), and alkaline phosphatase (ALP) activities were analyzed using kits from Jiangsu Yutong Biotechnology Co., Ltd. (Hangzhou, China) Additionally, serum catalase (CAT), malondialdehyde (MDA), glutathione peroxidase (GSH-PX), and total superoxide dismutase (SOD) levels and total antioxidant capacity (T-AOC) were measured.

#### 2.4.3. Meat Quality and Flavor Indices

After the feeding experiment, four fattening castrated boars with body weights close to the average were selected from each group, totaling 20 pigs. The live weight before slaughter was measured after a 12 h fasting period, followed by slaughter in accordance with the “Slaughter Operation Procedures for Pigs” (GB/T 17236-2008) (China, 2008). At 1 h and 24 h postmortem, the pH and color (L*, a*, and b* values) of the longissimus thoracis et lumborum muscle were determined by an electronic pH meter (model HI 9024C, HANNA Instruments, Ltd., Beijing, China) and Chroma meter (model CR-410 Konica Minolta Sensing Inc., Osaka, Japan) separately. The marbling score, shear force, cooking loss, drip loss, and water binding capacity of the longissimus thoracis et lumborum muscle were measured. Two longissimus thoracis et lumborum samples were taken from each pig: one for analyzing the fatty acid content and the other for analyzing the amino acid content. The content of inosine monophosphate (IMP) in the longissimus thoracis et lumborum muscle was measured with high-performance liquid chromatography (HPLC); for the fatty acid analysis, the freeze-dried longissimus thoracis et lumborum muscle samples were pulverized. The lipids of the pulverized samples were isolated with chloroform–methanol (1:1, *v*/*v*) in accordance with Folch [9]. The determination of fatty acid methyl esters was carried out on an Agilent 7890B gas chromatographer system with a flame ionization detector (Agilent Technologies Inc., Santa Clara, CA, USA). The composition of fatty acids in the longissimus thoracis et lumborum muscle was determined as previously described. Another method was conducted in accordance with the National Standard GB/T 39107-2020 (China, 2020) using Static Headspace Sampling (SHS) for extraction, with the analytical method employing the PEN3 electronic nose system (AIRSENSE Company, Schwerin, Germany). This e-nose system incorporates an array of 10 distinct metal oxide sensors (MOSs) to perform the detection and analysis of volatile flavor compounds(AIRSENSE Company, Schwerin, Germany). The sensor array enables the comprehensive characterization of volatile organic components through the pattern recognition of aroma profiles.

#### 2.4.4. Collection of Cecal Chyme and Determination of Intestinal Microbiota

Total genome DNA was extracted from colon samples from growing pigs using the QIAamp Fast DNA Stool mini kit (Qiagen, Hilden, Germany) and checked with 1% agarose gel. The DNA concentration and purity were determined with a Nano Drop 2000 UV-vis spectrophotometer (Thermo Fisher Scientific, Wilmington, NC, USA). The specific primer with the barcode 16S V3–V4 was amplified by an ABI Gene Amp R9700 PCR ther-mocycler (ABI, Los Angeles, CA, USA). Then, the PCR products were extracted, purified, and quantified. Paired-end sequencing was performed on an Illumina MiSeq PE300 platform/No-vaSeq PE250 platform (Illumina, San Diego, CA, USA). The raw 16S rRNA gene sequencing reads were demultiplexed, quality-filtered, and merged according to previous studies. The complexity of species diversity was evaluated with the ACE and Chao richness estimators and the diversity indices of Shannon and Simpson. The significant differences between samples were evaluated with the analysis of similarities (ANOSIM) (The web of majorbio). OTUs representing < 0.005% of the population were removed, and taxonomy was assigned using the RDP classifier (The web of majorbio). The relative abundance of each OTU was counted at different taxonomic levels. Then, bioinformatics analysis was mainly performed using QIIME (V1.7.0; San Diego, CA, USA) and R packages (Version 3.3.1, R Core Team, Vienna, Austria). The OTU table in QIIME was used to calculate the OTU level, and β diversity was assessed with principal coordinate analysis (PCoA). The cluster analysis and significant differences between samples were tested by ANOSIM. All procedures were conducted at Shanghai Meiji Biotechnology Co., Ltd. (Shanghai, China).

### 2.5. Data Analysis

All data were analyzed by one-way analysis of variance (ANOVA) using SPSS statistical software (Ver. 26.0 for Windows, SPSS, Inc., Chicago, IL, USA). Differences among treatments were examined using Tukey–Kramer multiple range tests, which were considered significant when the *p*-value was less than 0.05. The results are presented as means alongside their pooled standard errors of means (SEMs).

## 3. Results

### 3.1. Growth Performance

The growth performance data for Xiangling pigs fed diets supplemented with tea residue and complex enzymes are summarized in Table 3. There were no statistically significant differences in the initial body weight (IBW), final body weight (FBW), and average daily feed intake (ADFI) among the experimental groups (*p* > 0.05), indicating that the addition of tea residue alone did not adversely affect these parameters.

### 3.2. Serum Biochemical and Antioxidant Capacity

We analyzed the serum biochemical and antioxidant capacity of pigs in response to dietary tea residue and enzymes (Table 4). Dietary supplementation with tea residue considerably decreased the activity of AST and considerably increased the serum level of T-AOC and the CAT and SOD activities (*p* < 0.05). Tea residue supplementation significantly decreased the serum activity of ALP, which decreased to the lowest level in the M400 group (*p* < 0.05). Additionally, the incorporation of M400 into the diet remarkably reduced the contents of TC and HDL-C and the ALT activity in the serum and increased the serum level of T-AOC and the CAT and SOD activities in the serum (*p* < 0.05). In contrast, the level of serum MDA with dietary supplementation with complex enzymes was significantly lower than that in the control and tea residue groups (*p* < 0.05); no statistical difference was observed in TG among all groups (*p* > 0.05).

### 3.3. Meat Quality and Flavor Substances

We tested the meat quality (Table 5), muscle amino acids (Table 6), fatty acid profiles (Table 7), and volatile flavor substances (Table 8). Tea residue fed to pigs significantly decreased the pH_24h_ and b*1h in the longissimus dorsi muscle (*p* < 0.05). Additionally, the incorporation of M400 into the diet remarkably reduced the shear force and increased the L*_24h_ in the longissimus thoracis et lumborum muscle (*p* < 0.05). In addition, there were no significant differences in the pH_1h_, L*_1h_, a* (1 h and 24 h), b*_24h_, marbling score, cooking loss, drip loss, or water binding capacity among all groups (*p* > 0.05).

Tea residue fed to pigs increased the level of aspartic acid (*p* < 0.05). In addition, the content of glutamic acid, lysine, alanine, valine, tyrosine, proline, isoleucine, leucine, and phenylalanine in the longissimus thoracis et lumborum muscle was significantly increased in pigs fed with 200 mg/kg complex enzymes (*p* < 0.05). Moreover, neither tea residue nor complex enzymes fed to pigs affected the contents of threonine, serine, methionine, histidine, or arginine (*p* > 0.05).

Compared to the control group, there were no significant differences in the fatty acid content in the CZ group (*p* > 0.05). However, 800 mg/kg complex enzymes fed to pigs increased the C10:0, C15:0, and C17:0 contents (*p* < 0.05).

Tea residue supplementation significantly increased the inosine monophosphate content, which reached the highest level in the M400 group (*p* < 0.01). Tea residue significantly decreased alkane aroma (*p* < 0.05), while nitrogen oxides, short-chain alkanes, and alcohol compounds significantly decreased (*p* < 0.01), and ammonia and alkanes significantly increased (*p* < 0.01). Compared to the CZ group, the M200 group showed significantly increased alkane aromas, nitrogen oxides, short-chain alkanes, sulfur compounds, alcohols, and organic sulfur compounds (*p* < 0.01), while the ammonia, hydrogen, and alkane contents were significantly reduced (*p* < 0.01). Supplementing 400 mg/kg complex enzymes significantly increased alkane aromas and sulfur compounds, and the alkane content significantly decreased (*p* < 0.05).

### 3.4. Intestinal Microbiota

The microbial composition of the cecal digesta following tea residue and complex enzyme addition was revealed by 16S rRNA Illumina MiSeq sequencing. In the present study, the flattened rarefaction curves showed that the sampling in each group provided sufficient OTU coverage. The Venn diagram derived from Illumina MiSeq sequencing showed that the CON, CZ, M200, M400, and M800 groups contained 2319, 2663, 2589, 2352, and 2560 operational taxonomic units (OTUs), respectively, with 1116 OTUs shared among the three treatment groups. To assess the β diversity of the microbiota, the Bray method was employed to calculate the distance among all OTUs, followed by principal coordinate analysis (PCoA). The PCoA results revealed that samples within each treatment group clustered closely, indicating similar species compositions and minimal differences between groups. The 95% confidence intervals for the CON, CZ, and M200 groups overlapped, suggesting their bacterial microbiota compositions were relatively alike (Figure 1). No significant differences in species richness (as reflected by the ACE and Chao1 indices) or diversity (as reflected by the Shannon and Simpson indices) were observed in the cecal digesta bacteria at the taxonomic level (Figure 2).

At the phylum level, *Firmicutes* and *Bacteroidetes* were the two dominant phyla, contributing 76.31% and 19.64% to the control group, 82.54% and 12.85% to the CZ group, 87.84% and 7.40% to the M200 group, 79.33% and 11.71% to the M400 group, and 83.78% and 13.25% to the M800 group, respectively. *Spirochaetota* and *Verrucomicrobiota* were the next two most dominant phyla, accounting for 1.78% and 1.09% in the control group, 1.45% and 1.18% in the CZ group, 1.94% and 1.23% in the M200 group, 3.62% and 0.99% in the M400 group, and 0.64% and 0.85% in the M800 group (Figure 3). No significant changes were found in the abundances of any of the phyla. The CZ diet induced a significant enrichment of *Oribacterium* and *Butyricicoccus*, the M200 diet induced a significant enrichment of *Streptococcaceae*, *Streptococcus*, and *Coriobacteriales*, and the M400 diet induced a significant enrichment of *Eggerthellaceae*, *Oscillospirales*, and *Peptococcacae* (Figure 4). Correlation analysis between differential microorganisms and serum biochemistry (Figure 5) revealed a significant negative correlation between *Peptococcacae* and AST in blood lipid metabolism (*p* < 0.05). Furthermore, *Erysipelotrichales* demonstrated an extremely significant correlation with HDL-C and ALP (*p* < 0.01), while *Eggerthellaceae* showed a significant negative correlation with HDL-C (*p* < 0.05).

## 4. Discussion

### 4.1. Effects of Tea Residue and Complex Enzymes on the Growth Performance of Xiangling Fattening Pigs

Most previous studies focused on the combination of other by-product meals and compound enzymes, while limited research has addressed the synergistic effects of tea residue and complex enzymes in pig diets. In our study, although tea residue inclusion did not significantly affect growth performance, this observation aligns with the trend reported by Gou [10], who found that adding 6% tea residue reduced the growth performance of fattening pigs. The potential explanation for this lies in the presence of anti-nutritional factors such as tannins and caffeine in tea residue, which can form complexes with proteins and digestive enzymes, thereby impairing nutrient absorption and metabolic efficiency [11]. Complex enzyme supplementation, particularly at moderate levels (M200 group), may have partially alleviated these effects by promoting fiber degradation and hydrolyzing tannin–protein complexes, thus potentially improving nutrient availability. However, as our data did not reveal significant differences among treatments, the impact of enzyme levels on growth performance requires further investigation. These findings suggest that while enzyme addition could help mitigate the anti-nutritional effects of tea residue, the optimal dosage should be carefully determined to maximize potential benefits without compromising dietary balance.

### 4.2. Effects of Tea Residue and Complex Enzyme Addition on Serum Biochemical and Antioxidant Indicators of Xiangling Fattening Pigs

Serum biochemical indicators are essential for evaluating animal health and metabolic status [12]. In this study, the inclusion of tea residue significantly reduced AST activity, indicating a potential improvement in liver health. However, ALT activity was not affected by tea residue alone and showed a significant decrease only in combination with complex enzyme supplementation (M400 group). Previous research has shown that catechins in tea residue can lower the TC levels in goats [13,14] and regulate lipid metabolism [15]. Consistent with this, we observed a reduction in the serum TC levels in the M400 group. Notably, AST and ALT are primarily indicators of liver function rather than direct markers of lipid metabolism; thus, changes in these enzymes reflect hepatic status rather than lipid metabolic pathways. Tea polyphenols possess antioxidant properties, capable of scavenging free radicals and enhancing endogenous antioxidant enzymes such as GSH-PX and SOD, thereby protecting tissues from oxidative damage. The decrease in serum TC in the M400 group may be attributed to the combined effects of tea polyphenols and the action of complex enzymes, including cellulase, xylanase, alkaline protease, and tannase, which potentially improved nutrient utilization and antioxidant defense [16].

The T-AOC, SOD, CAT, GSH-PX, and MDA are important indicators of antioxidant capacity [17]. Catechins have been reported to lower the MDA levels in mouse serum [18,19]. Similarly, our study found that both tea residue and complex enzyme supplementation reduced the serum MDA levels. The improvement in the T-AOC with tea residue supplementation aligns with previous findings [20]. CAT activity was significantly increased in the CZ and M400 groups, consistent with earlier reports [21,22]. Moreover, while tea residue alone increased the SOD levels but decreased GSH-PX activity, complex enzyme supplementation, particularly in the M400 group, significantly enhanced the SOD levels [23,24]. These results indicate that the combined use of tea residue and complex enzymes improved the antioxidant capacity of fattening pigs, with the M400 group showing the most notable effects.

### 4.3. Effects of Tea Residue and Complex Enzyme Addition on the Quality and Flavor Substances of Xiangling Fattening Pork

Meat quality is a major factor influencing consumer acceptance, as it determines the tenderness, juiciness, and flavor of pork [25]. In our study, pigs fed the M400 diet showed a significant reduction in shear force compared with the control group, indicating improved meat tenderness. However, changes in the L24h and pH24h values observed in this group require cautious interpretation, as their impacts on meat quality are complex and may vary among different pig breeds [26,27]. Previous research reported that the addition of 4% tea residue reduced pork shear force [28]; in our study, this effect was only observed when tea residue was combined with complex enzymes. Meat color and water retention are important indicators of pork quality, but the L value standards primarily apply to commercial pig breeds and may not directly translate to Xiangling pigs [29]. Overall, the reduction in shear force suggests that the M400 diet improved the tenderness of Xiangling pork.

The amino acid profile in muscle is a key determinant of protein quality and meat flavor [30]. In our study, pigs fed the M200 diet showed elevated levels of glutamic acid, lysine, alanine, valine, tyrosine, proline, isoleucine, leucine, and phenylalanine. These amino acids serve as important precursors for flavor compounds, participating in reactions that enhance the umami and aromatic characteristics of meat [31]. Notably, glutamic acid and inosine monophosphate (IMP) are well recognized for their umami contribution [32]. Tea residue supplementation significantly increased the IMP content in muscle, which could potentially enhance the umami intensity of pork. Additionally, amino acids such as alanine, glycine, lysine, and proline contribute to sweetness, while valine, leucine, and isoleucine impart mild bitterness, contributing to overall flavor complexity [33,34].

The fatty acid composition is another crucial factor influencing meat quality [35]. Unsaturated fatty acids undergo lipid oxidation during heating, generating volatile flavor compounds and contributing to meat aroma development [36,37]. Moreover, sulfur-containing amino acids such as methionine and cysteine play a pivotal role in forming aromatic compounds, enriching the characteristic flavor of pork [38]. Previous studies reported that tea residue supplementation increased the amino acid, unsaturated fatty acid, and IMP contents in mutton [28], while Wu [39] observed elevated C16:0 and C20:1 fatty acid levels in tea-residue-fed chickens, although differences in species and diets make direct comparison challenging. These effects could be attributed to the bioactive compounds in tea residue, such as theanine, which may influence protein and lipid metabolism, thereby affecting flavor precursors. Furthermore, complex enzyme supplementation appeared to promote the release and absorption of these compounds, as evidenced by increased levels of certain amino acids and IMP, particularly in the M200 and M400 groups. These findings indicate that the combination of tea residue and moderate enzyme supplementation may help improve the nutritional profile and flavor precursors of Xiangling pork.

### 4.4. Effects of Tea Residue and Complex Enzyme Addition on the Intestinal Microbiota of Xiangling Fattening Pigs

The intestinal microbiota functions as a dynamic ecosystem influenced by various environmental factors, including diet, age, and individual genotype. Tea residue is rich in cellulose and polyphenols, which can interact with intestinal microorganisms [40,41]. Previous studies reported that increased tea residue in the diet corresponded with an increase in beneficial bacterial genera [42], consistent with our findings. In our study, tea residue supplementation enriched microbial diversity, with Oribacterium identified as a dominant genus. Research has indicated that Oribacterium abundance is negatively correlated with pneumonia inflammation [43], suggesting a potential health benefit, although the precise mechanisms remain to be clarified.

Additionally, studies have shown that incorporating complex enzyme preparations into pig diets can enhance microbial diversity [44]. Similar findings have reported that complex enzymes reduce harmful bacteria while increasing beneficial genera [45]. In our study, Erysipelotrichales was the dominant bacterial group in the control pigs. Previous research has associated a higher abundance of Erysipelotrichales with an increased risk of fatty liver disease [46]. Although the control group exhibited elevated ALP levels, which could suggest alterations in lipid metabolism, our study did not directly assess the liver health status, and further investigation is needed to establish this connection. In the M200 group, Streptococcaceae emerged as the dominant bacterial group. This genus is commonly present in the nasopharynx and gastrointestinal tract, with many strains regarded as part of the normal microbiota. In contrast, Oscillospirales was dominant in the M400 group. Oscillospirales belongs to the Ruminococcaceae family within the Clostridiales order and has been associated with fiber degradation and beneficial metabolic functions [47,48,49,50]. Although the specific functional roles of these microbial shifts require further clarification, our findings suggest that the dietary inclusion of tea residue combined with complex enzymes, particularly at the M400 level, may help modulate the gut microbial composition toward a more favorable profile in fattening pigs.

## 5. Conclusions

In conclusion, the inclusion of tea residue alone did not significantly affect the growth performance of Xiangling fattening pigs. However, supplementing tea-residue-based diets with complex enzymes improved certain physiological parameters, including the serum antioxidant capacity and meat tenderness, as evidenced by the reduced shear force in the M400 group. Notably, the 400 mg/kg enzyme supplementation group (M400) exhibited the most pronounced improvements in antioxidant indicators and meat quality traits. Although the intestinal microbiota diversity was not significantly altered, enzyme supplementation contributed to beneficial shifts in the microbial composition, such as an increased abundance of Oscillospirales. Overall, this study provides evidence that combining tea residue with an appropriate level of complex enzymes, particularly at 400 mg/kg, may help improve the antioxidant status and some meat quality parameters in Xiangling pigs. Further research is warranted to optimize enzyme dosages, investigate nutrient digestibility, and explore the potential sensory impacts and long-term production outcomes of this dietary strategy.

## Figures and Tables

**Figure 1 animals-15-01229-f001:**
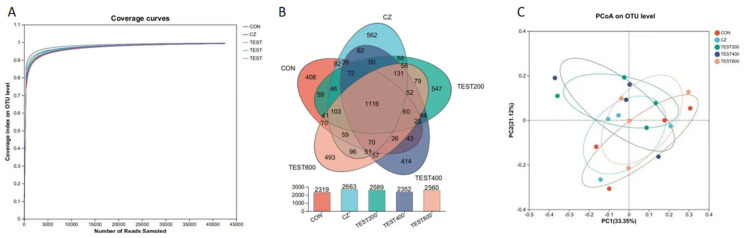
Effects of tea residue and different levels of complex enzymes on the diversity of pig intestinal microorganisms (*n* = 4) (**A**) Dilution curve; (**B**) Venn diagram of OTUs; (**C**) PCoA analysis.

**Figure 2 animals-15-01229-f002:**
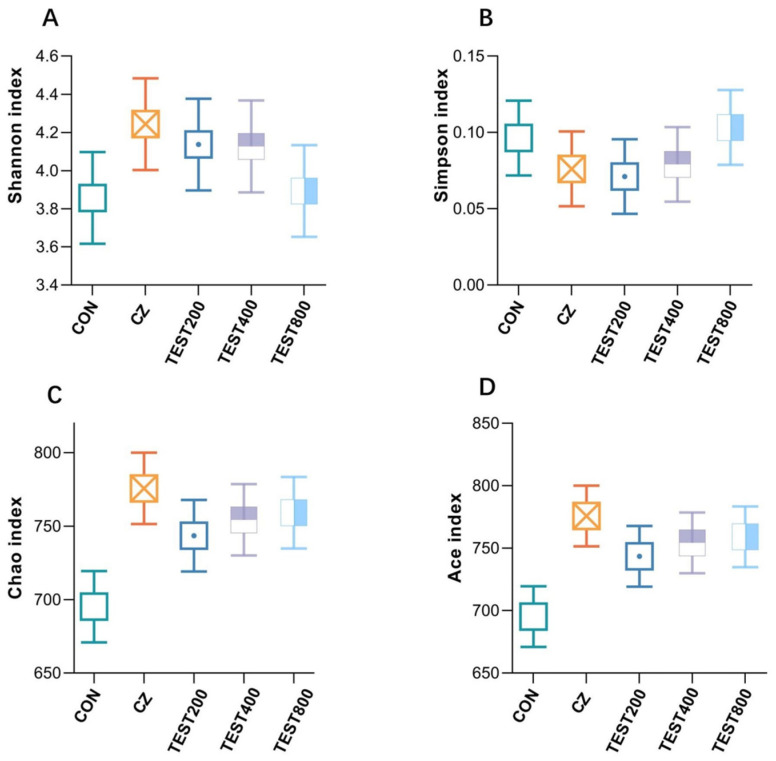
Effects of tea residue and different levels of complex enzymes on the diversity of pig intestinal microorganisms (*n* = 4) (**A**–**D**): α diversity analysis.

**Figure 3 animals-15-01229-f003:**
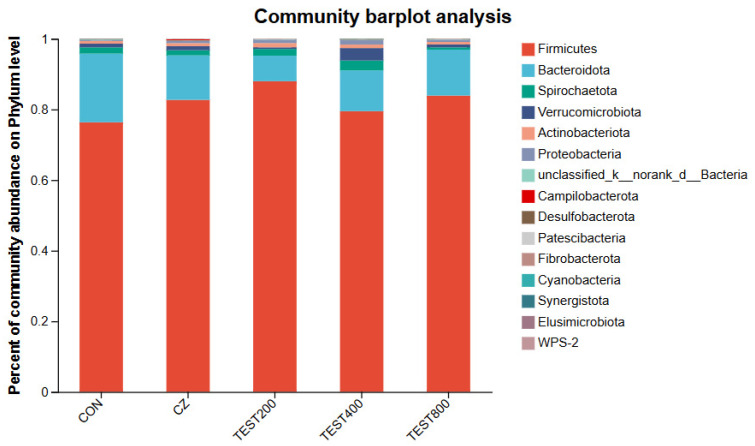
Effects of tea residue and different levels of complex enzymes on the composition of pig intestinal microorganisms (phylum level, *n* = 4).

**Figure 4 animals-15-01229-f004:**
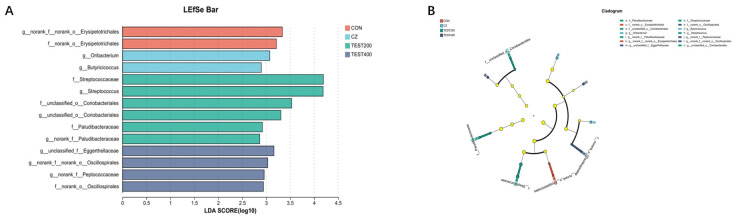
Effects of tea residue and different levels of complex enzymes on LefSE analysis of cecal contents (**A**) and LefSE analysis of Cladogram (**B**) in finishing pigs (genus level, *n* = 4).

**Figure 5 animals-15-01229-f005:**
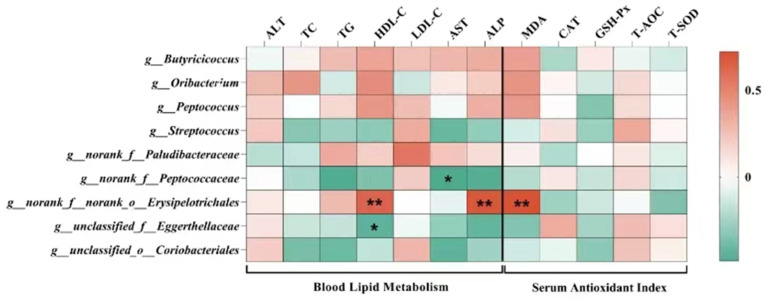
Correlation analysis between differential microorganisms and blood metabolites and antioxidant levels (*, *p* < 0.05; **, *p* < 0.01).

**Table 1 animals-15-01229-t001:** Nutrients and anti-nutritional factors in tea residue (air-dried basis).

Items	Tea Residue
Gross energy (MJ/kg)	21.43
Dry matter %	92.32
Crude protein %	28.12
Crude fiber %	17.56
Neutral detergent fiber %	58.74
Acid detergent fiber %	32.51
Crude ash %	4.30
Calcium %	0.55
Total phosphorus %	0.22
Tannin %	0.08
Caffeine %	0.50
Lysine %	1.89
Methionine %	1.53
Threonine %	0.05
Tryptophan %	0.32

**Table 2 animals-15-01229-t002:** Composition and nutrient levels of the basal diet (air-dried basis).

Ingredient %	Basal Diet	Tea Residue Feed
Tea residue		5.80
Paddy	49.04	52.00
Corn	7.00	
Wheat bran	6.00	11.70
Rice bran	15.00	8.50
Defatted rice bran	3.00	
Sugar cane molasses	1.50	1.50
Broken rice	10.00	14.00
43 soybean meal	5.50	3.70
Salt	0.40	0.41
Limestone	1.05	0.96
Dicalcium phosphate	0.16	0.20
70% lysine	0.54	0.46
98% threonine	0.05	0.01
Choline chloride	0.05	0.05
Antioxidant	0.01	0.01
Antifungal agents	0.20	0.20
Premix	0.50	0.50
Total	100.00	100.00
Calculated Nutritional Value		
Digestive energy MJ/Kg	11.97	11.98
Crude protein %	11.26	11.49
Calcium %	0.54	0.54
Total phosphorus %	0.60	0.57
Energy-to-nitrogen ratio	1.06	1.05
Lysine %	0.79	0.79
Methionine %	0.22	0.22
Threonine %	0.44	0.44
Tryptophan %	0.12	0.11

The vitamin and mineral premix contained the following per kg of diet: VA, 4200 IU; VD, 3400 IU; VE, 36 IU; VK_3_, 1.2 mg; VB_12_, 23 μg; VB_2_, 5.63 mg; VB_5_, 20.5 mg; VB_3_, 28 mg; choline chloride, 1.00 g; folic acid, 0.8 mg; VB_1_, 3.4 mg; VB_6_, 2.7 mg; VH, 0.18 mg; Mn (as manganese sulfate), 40.0 mg; Fe (as ferrous sulfate), 70.0 mg; Zn (as zinc sulfate), 70.0 mg; Cu (as copper sulfate), 70 mg; I (as potassium iodide), 0.3 mg; and Se (as sodium selenite), 0.3 mg.

**Table 3 animals-15-01229-t003:** Effect of tea residue and complex enzyme addition on growth performance of fattening pigs.

Items	CON	CZ	M200	M400	M800	SEM	*p*-Value
IBW (kg)	48.00	48.01	47.86	47.87	47.94	0.62	0.86
FBW (kg)	86.43	83.51	85.32	83.15	81.99	0.89	0.27
TWG (kg)	37.9	35.5	36.9	35.3	34.1	0.66	0.4
ADFI (g/d)	1777.29	1784.06	1675.82	1785.15	1779.94	0.83	0.41
FCR	4.01	4.23	3.91	4.23	4.48	0.64	0.4

IBW, initial body weight; FBW, final body weight; TWG, total weight gain; ADFI, average daily feed intake; FCR, feed conversion ratio.

**Table 4 animals-15-01229-t004:** Effect of tea residue and complex enzyme addition on serum biochemical and antioxidant capacity of fattening pigs.

Items	CON	CZ	M200	M400	M800	SEM	*p*-Value
TC/(mmol/L)	2.70 ^ab^	2.74 ^a^	2.69 ^ab^	2.61 ^b^	2.71 ^ab^	0.020	0.03
TG/(mmol/L)	0.56	0.54	0.55	0.54	0.55	0.004	0.93
HDL-C/(mmol/L)	0.40 ^a^	0.39 ^ab^	0.38 ^ab^	0.37 ^b^	0.39 ^ab^	0.003	0.01
LDL-C/(mmol/L)	1.67 ^ab^	1.68 ^a^	1.69 ^a^	1.65 ^ab^	1.60 ^b^	0.010	0.02
ALP/(U/L)	190.79 ^a^	182.12 ^b^	184.11 ^b^	180.56 ^b^	182.27 ^b^	1.130	<0.01
ALT/(U/L)	20.27 ^a^	19.68 ^ab^	20.14 ^ab^	19.10 ^b^	19.38 ^b^	1.390	<0.01
AST/(U/L)	41.40 ^a^	39.01 ^b^	38.87 ^b^	38.33 ^b^	40.38 ^ab^	0.350	0.02
T-AOC/(U/mL)	7.33 ^b^	7.75 ^a^	7.45 ^b^	7.86 ^a^	7.50 ^b^	0.060	0.01
MDA/(nmol/L)	10.18 ^a^	10.03 ^a^	9.75 ^b^	9.64 ^b^	9.83 ^b^	0.050	<0.01
GSH-PX/(U/mL)	189.81 ^ab^	178.76 ^b^	186.98 ^b^	186.55 ^b^	195.74 ^a^	1.460	<0.01
CAT/(U/mL)	1.19 ^b^	1.30 ^a^	1.16 ^b^	1.26 ^a^	1.21 ^b^	0.011	<0.01
SOD/(U/mL)	1.88 ^b^	1.96 ^a^	1.95 ^ab^	2.00 ^a^	1.92 ^ab^	0.013	<0.01

Note: SEM—Standard error of mean value; *p* < 0.05 indicates a statistical difference, specifically referring to the shoulder letters of peer data.TC, total cholesterol; TG, triglyceride; HDL-C, high-density lipoprotein cholesterol; LDL-C, low-density lipoprotein cholesterol; ALP, alkaline phosphatase; ALT, alanine aminotransferase; AST, aspartate aminotransferase; T-AOC, total antioxidant capacity; MDA, malondialdehyde; GSH-PX, glutathione peroxidase; CAT, catalase; SOD, superoxide dismutase.

**Table 5 animals-15-01229-t005:** Effect of tea residue and complex enzyme addition on meat quality of fattening pigs.

Items	CON	CZ	M200	M400	M800	SEM	*p*-Value
pH_1h_	6.42	6.45	6.26	6.35	6.26	0.059	0.81
pH_24h_	6.04 ^a^	5.77 ^b^	5.72 ^b^	5.62 ^b^	5.72 ^b^	0.046	0.02
L*_1h_	39.07	38.4	40.5	37.23	39.13	0.473	0.3
L*_24h_	41.12 ^b^	43.18 ^ab^	43.15 ^ab^	45.33 ^a^	42.73 ^ab^	0.630	0.04
a*_1h_	7.6	6.32	6.61	6.29	6.76	0.311	0.72
a*_24h_	7.81	7.58	8.63	9.11	8.48	0.313	0.57
b*_1h_	2.72 ^a^	1.84 ^b^	2.43 ^ab^	1.98 ^ab^	2.08 ^ab^	0.126	0.02
b*_24h_	3.98	4.81	5.75	5.91	4.85	0.251	0.26
Marbling score	3.375	3.375	3.25	3.125	3.25	0.057	0.66
Shear force (N)	12.97 ^a^	12.46 ^a^	10.98 ^ab^	9.29 ^b^	11.21 ^ab^	0.469	< 0.01
Cooking loss (%)	0.72	0.7	0.73	0.71	0.65	0.114	0.08
Drip loss (%)	0.018	0.015	0.015	0.016	0.015	0.001	0.08
Water binding capacity (%)	0.868	0.863	0.851	0.837	0.847	0.015	0.26

Note: SEM—Standard error of mean value; *p* < 0.05 indicates a statistical difference, specifically referring to the shoulder letters of peer data.

**Table 6 animals-15-01229-t006:** Effect of tea residue and complex enzyme addition on muscle amino acid composition of fattening pigs (%).

Items	CON	CZ	M200	M400	M800	SEM	*p*-Value
Aspartic acid	2.05 ^b^	2.07 ^a^	2.13 ^a^	2.07 ^a^	2.07 ^a^	0.011	0.02
Threonine	1.02	1.03	1.05	1.03	1.03	0.005	0.32
Serine	0.87	0.87	0.9	0.87	0.88	0.005	0.29
Glutamic acid	3.25 ^b^	3.24 ^b^	3.36 ^a^	3.26 ^b^	3.26 ^b^	0.016	0.03
Glycine	0.94 ^ab^	0.95 ^ab^	0.96 ^a^	0.93 ^b^	0.95 ^ab^	0.004	0.02
Alanine	1.27 ^b^	1.28 ^ab^	1.31 ^a^	1.28 ^ab^	1.29 ^ab^	0.006	0.02
Methionine	0.34	0.35	0.36	0.36	0.34	0.005	0.73
Valine	1.15 ^b^	1.16 ^b^	1.20 ^a^	1.17 ^ab^	1.17 ^ab^	0.007	0.02
Isoleucine	1.00 ^b^	1.01 ^b^	1.07 ^a^	1.02 ^b^	1.02 ^b^	0.007	<0.01
Leucine	1.77 ^b^	1.78 ^b^	1.86 ^a^	1.79 ^b^	1.79 ^b^	0.010	<0.01
Tyrosine	0.60 ^b^	0.61 ^ab^	0.63 ^a^	0.60 ^b^	0.60 ^b^	0.004	0.02
Lysine	1.96 ^b^	1.98 ^b^	2.04 ^a^	1.99 ^ab^	1.98 ^b^	0.010	0.01
Phenylalanine	0.89 ^b^	0.90 ^b^	0.94 ^a^	0.91 ^ab^	0.91 ^ab^	0.023	<0.01
Histidine	1.01	1.04	1.07	1.01	1.06	0.012	0.29
Arginine	1.37	1.38	1.41	1.37	1.37	0.006	0.26
Praline	0.85 ^ab^	0.84 ^b^	0.88 ^a^	0.83 ^b^	0.83 ^b^	0.006	<0.01

Note: SEM—Standard error of mean value; *p* < 0.05 indicates a statistical difference, specifically referring to the shoulder letters of peer data.

**Table 7 animals-15-01229-t007:** Effect of tea residue and complex enzyme addition on muscle fatty acid composition of fattening pigs (%).

Items	CON	CZ	M200	M400	M800	SEM	*p*-Value
Capric acid (C10:0)	0.06 ^b^	0.07 ^ab^	0.06 ^b^	0.08 ^ab^	0.09 ^a^	0.004	0.03
Lauric acid (C12:0)	0.06	0.07	0.07	0.06	0.07	0.004	0.92
Myristic acid (C14:0)	1.11	1.22	1.09	1.13	1.1	0.034	0.84
Pentadecanoic acid (C15:0)	0.04 ^b^	0.05 ^ab^	0.05 ^ab^	0.05 ^ab^	0.07 ^a^	0.009	0.02
Palmitic acid (C16:0)	22.95 ^ab^	23.48 ^a^	22.68 ^ab^	22.98 ^ab^	21.85 ^b^	0.229	0.03
Palmitoleic acid (C16:1)	3.27	3.41	3.36	3.83	3.78	0.125	0.54
Margaric acid (C17:0)	0.12 ^b^	0.15 ^ab^	0.14 ^ab^	0.14 ^ab^	0.17 ^a^	0.007	0.02
Heptadecenoic acid (C17:1)	0.11	0.12	0.12	0.12	0.14	0.005	0.43
Stearic acid (C18:0)	11.63	11.45	11.58	11.6	11.23	0.188	0.37
Oleic acid (C18:1n-9c)	41.3	40.85	41.7	40.68	38.1	0.837	0.73
Linoleic acid (C18:2n-6c)	12.82	11.69	12.06	12.49	14.8	0.668	0.67
α-Linolenic acid (C18:3n-3)	0.29	0.31	0.3	0.32	0.35	0.013	0.59
Arachidic acid (C20:0)	0.25	0.26	0.26	0.24	0.25	0.005	0.48
Eicosenoic acid (C20:1)	0.84 ^ab^	0.82 ^ab^	0.88 ^a^	0.76 ^ab^	0.68 ^b^	0.026	0.04
Eicosadienoic acid (C20:2)	0.52	0.53	0.56	0.55	0.64	0.028	0.72
Eicosenoic acid (C20:3n-6)	0.38	0.4	0.41	0.41	0.54	0.029	0.43
Eicosenoic acid (C20:3n-3)	0.06	0.08	0.09	0.1	0.11	0.007	0.32
Eicosenoic acid (C20:4n-6)	3.56	3.29	3.84	3.79	4.89	0.297	0.54
Heneicosanoic acid (C21:0)	0.11	0.12	0.12	0.14	0.17	0.010	0.47
Behenic acid (C22:0)	0.18	0.23	0.25	0.19	0.36	0.026	0.72
Tetracosanoic acid (C24:0)	0.14	0.12	0.11	0.12	0.17	0.010	0.42
Nervonic acid (C24:1)	0.23	0.25	0.29	0.21	0.47	0.039	0.19

Note: SEM—Standard error of mean value; *p* < 0.05 indicates a statistical difference, specifically referring to the shoulder letters of peer data.

**Table 8 animals-15-01229-t008:** Effect of tea residue and complex enzyme addition on muscle flavor substance components of fattening pigs (%).

Items	CON	CZ	M200	M400	M800	SEM	*p*-Value
Inosine monophosphate	1.75 ^b^	2.10 ^a^	2.15 ^a^	2.18 ^a^	2.14 ^a^	0.065	<0.01
Aromatic ingredients	0.994 ^b^	0.997 ^ab^	0.999 ^a^	1.001 ^a^	1.000 ^a^	0.014	0.02
Nitrogen oxides	1.349 ^a^	1.245 ^b^	1.368 ^a^	1.249 ^b^	1.256 ^b^	0.003	<0.01
Ammonia	0.954 ^b^	0.959 ^a^	0.956 ^b^	0.961 ^a^	0.959 ^a^	0.001	<0.01
Hydrogen	1.001 ^a^	1.002 ^a^	0.999 ^b^	1.003 ^a^	1.002 ^a^	0.001	<0.01
Alkane aromatic ingredients	0.974 ^b^	0.973 ^b^	0.975 ^a^	0.975 ^a^	0.972 ^b^	0.001	0.03
Short-chain alkanes	2.834 ^a^	2.303 ^b^	2.745 ^a^	2.274 ^b^	2.312 ^b^	0.027	<0.01
Sulfides	1.093 ^b^	1.104 ^b^	1.177 ^a^	1.136 ^a^	1.112 ^b^	0.007	<0.01
Alcohols	1.379 ^a^	1.249 ^b^	1.363 ^a^	1.238 ^b^	1.255 ^b^	0.007	<0.01
Organic sulfides	1.511 ^b^	1.416 ^b^	1.601 ^a^	1.435 ^b^	1.442 ^b^	0.007	<0.01
Alkanes	1.044 ^b^	1.046 ^a^	1.043 ^b^	1.043 ^b^	1.045 ^a^	0.001	<0.01

Note: SEM—Standard error of mean value; *p* < 0.05 indicates a statistical difference, specifically referring to the shoulder letters of peer data.

## Data Availability

Data are contained within this article.

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
