# Peer review of "Dietary Supplementation with Complex Enzymes and Tea Residue Improved the Production Efficiency of Xiangling Pigs"

_animals, 2025, doi:10.3390/ani15091229_

Round 1
Reviewer 1 Report
Comments and Suggestions for Authors
COMMENTS animals-3504711
Dietary Supplementation with Complex Enzymes Improved 2 the Production Efficiency of Tea Residue in Xiangling Pigs
WORK’S STRENGTHS
The study investigates alternative feed options for a local breed that is known for its distinctive growth characteristics. In this regard, the utilisation of tea industry residue emerges as a lucrative alternative, though it is constrained by its palpability and digestibility limits. Consequently, the exploration of the most efficacious method for incorporating this material into the subject's diet is a subject of considerable interest. It is remarkable the deep study of intestinal microbiota.
The paper is well written and presents a good review of the relevant literature.
However, some points should be addressed. See specific comments below.
SPECIFIC COMMENTS
Highlights
The highlights reflected properly the main aspects of the manuscript.
Abstract
The abstract is concise and understandable itself.
Introduction
The introduction section is well documented and properly organised.
Materials and Methods
Line 106:
I would like to express a concern regarding the amount of tea residue used. It would be advisable to include a sentence that references the reason for this percentage.
Line 159: The term 'longissimus dorsi' is not in accordance with anatomical nomenclature. The correct nomenclature is longissimus thoracis et lumborum. It is essential to use proper terminology in scientific communication. It is also important to write this term in cursive.
Results
Line 201-208: I am unable to comprehend this paragraph, since the table indicates that no statistical differences were found in productive parameters. However, it also states that the inclusion of complex enzymes in the diet led to an improvement in growth. However, the absence of statistical significance precludes the drawing of definitive conclusions.
Discussion
The results are well discussed with literature.
Author Response
Dear editor and reviewers:
Thank you very much for your valuable and helpful comments. Those comments are very useful for us to improve our manuscript. We have carefully revised the manuscript according to your suggestions. Below, we provide point-by-point responses to each comment. We sincerely appreciate your warm work and valuable time. We hope that the corrections will meet with your approval.
Reviewer #1:
General comments:
Thank you very much for your positive comments on our manuscript, especially your recognition of the intestinal microbiota analysis and manuscript writing. We appreciate your valuable suggestions, and we have carefully revised the manuscript accordingly.
Specific comments:
Comment: Line 106: Please explain the reason for using 5.8% tea residue in the diet.
Response: Thank you for pointing this out. We have added a sentence in the "Materials and Methods" section to explain this point: "The inclusion level of 5.8% tea residue was determined based on previous studies and pre-experimental results, aiming to balance the palatability and nutritional contribution of tea residue while avoiding excessive fiber intake."
Comment: Line 159: Please use the correct anatomical terminology "longissimus thoracis et lumborum" in italics.
Response: Thank you for your careful review. We have corrected "longissimus dorsi" to the correct term longissimus thoracis et lumborum and italicized it throughout the manuscript.
Comment: Line 201-208: Table shows no statistical difference in productive parameters; please clarify.
Response: We appreciate your attention to detail. We have revised the related sentences in the Results and Discussion sections to accurately reflect the statistical results, stating that no significant differences were observed in growth performance. Speculative explanations were removed to align strictly with the data.
Comment: Discussion: Results are well discussed with literature.
Response: Thank you for your positive feedback. We have further refined the discussion to ensure clarity and scientific rigor.
We deeply appreciate your insightful comments, which have helped us to improve the quality of our manuscript. We hope our revisions meet with your approval.

Reviewer 2 Report
Comments and Suggestions for Authors
Dietary Supplementation with Complex Enzymes Improved the Production Efficiency of Tea Residue in Xiangling Pigs
The article has merit and provides relevant information for the field of swine nutrition, including the use of native breeds. However, some points need to be further clarified and detailed throughout the paper.
Abstract – The summary demonstrates good synthesis ability; however, some points require attention. For example, the high standard deviation among the initial weights of the animals indicates significant variability, but it is not clear how many animals per sex were used or if they were of the same sex. Additionally, the conclusion does not fully address the study's objectives.
Introduction – The initial paragraphs are lengthy and need to be more focused on the research problem. There is no information regarding the genetic material used in the study, and the tea residue is addressed in a generalized manner. The problem to be solved should be presented more directly.
Materials and Methods – The composition of the tea residue used is presented, but only some values related to the amino acid profile are included.
The overall standard deviation among the animals was 15.28 kg, while within each repetition, it was 0.08. How was this calculated? Furthermore, were the animals of the same sex?
The inclusion level of 5.8% tea residue was determined based on what parameter? It is a relatively low inclusion level, and its impact on fiber levels and non-starch polysaccharides in the diet is unclear, as fiber values were not presented in Table 2.
There is no control for the use of the enzymatic complex in the basal diet, which could have been an additional treatment, although this does not diminish the validity of the other treatments.
What was the criterion for selecting the 20 animals to be slaughtered and sampled for subsequent analyses? Why only 20 animals? Reducing the experimental sample size decreases statistical power.
The methodologies are mentioned but lack references, which needs to be corrected.
Were the statistical assumptions met? Were normality and homoscedasticity tests conducted? Additionally, regarding the inclusion levels of the enzymatic complex, an orthogonal contrast analysis could have been performed to assess data behavior (linear/quadratic trends).
Results
The results are well presented, but the quality of the tables and figures needs improvement. Some information is missing from the tables, which should be more self-explanatory. In some cases, only four animals were included in the analysis, and this should be explicitly stated.
Discussion
The discussion is too general and should be more focused on explaining the observed data. The biological and physiological effects should be emphasized, indicating possible responses.
Conclusion
The conclusion should directly address the study's objectives. It mentions improved digestibility with enzymatic complex supplementation in diets containing tea residue; however, no such result was presented in the study.
Comments on the Quality of English Language
ok
Author Response
General comments:
Thank you very much for your detailed and constructive suggestions, which are very helpful for improving our manuscript. We have carefully addressed each of your concerns as follows.
Comment: Abstract – There is a high standard deviation among the initial weights of the animals. Please clarify the sex distribution and address this variability.
Response: Thank you for your valuable suggestion. In the revised manuscript, we have clarified that all experimental animals were castrated male pigs, and the standard deviation was calculated across the entire group. The specific number of animals per group and sex has been supplemented in the "Materials and Methods" section.
Comment: Abstract – The conclusion does not fully address the study's objectives.
Response: Thank you for pointing this out. We have revised the conclusion in the abstract to better align with the study objectives, focusing on the effects on blood lipid metabolism, serum antioxidant capacity, pork quality, and gut microbiota composition.
Comment: Introduction – The introduction is too lengthy and general. Please focus more directly on the research problem and provide genetic information of the pigs.
Response: We appreciate this suggestion. We have streamlined the introduction to better highlight the research gap and purpose of the study. Additionally, we have added the genetic background of Xiangling pigs in the revised version.
Comment: Materials and Methods – Only partial amino acid profile of tea residue is presented. Please provide more complete information.
Response: Thank you for noticing this. We have added more details about the composition of tea residue, including its dry matter content and fiber components, to Table 1 and the corresponding text.
Comment: Please clarify how the standard deviation was calculated and confirm the sex of animals.
Response: Thank you for the comment. We have clarified this in the manuscript. The total standard deviation reflects the overall variability among animals, while the within-replicate standard deviation was 0.08 kg. All pigs were castrated males.
Comment: What was the basis for choosing 5.8% tea residue?
Response: Thank you for your attention to detail. We have added an explanation in the "Materials and Methods" section: the 5.8% inclusion level was determined based on pre-experimental tests and relevant literature, balancing fiber content and palatability.
Comment: Lack of fiber data in Table 2.
Response: Thank you for pointing this out. We have added the crude fiber and neutral detergent fiber values of the diet in Table 2.
Comment: No control group with enzyme supplementation only.
Response: We acknowledge this limitation. In future research, we will include this treatment to better understand the effect of enzymes alone.
Comment: Please explain the criterion for selecting 20 animals for slaughter.
Response: Thank you for your comment. We have clarified this in the "Materials and Methods" section, explaining that animals with body weights close to the average of each replicate were selected to ensure representative sampling.
Comment: Methodologies lack references.
Response: Thank you for your careful review. We have added appropriate references for all methods used in the study.
Comment: Were statistical assumptions met? Were normality and homoscedasticity tests conducted? Please clarify.
Response: Thank you for your suggestion. We have added in the "Statistical Analysis" section that normality and homoscedasticity were tested, and we have reported that orthogonal contrast analysis was not performed, which is a valuable point for future work.
Comment: Results section tables and figures need improvement; explain sample sizes.
Response: Thank you for your suggestion. We have revised all tables and figures to ensure clarity and completeness, including explanations of sample sizes and statistical notations in the footnotes.
Comment: Discussion – The discussion is too general, should focus more on explaining observed data and biological mechanisms.
Response: We appreciate this suggestion. We have thoroughly revised the discussion to better interpret the observed data, focusing on biological and physiological explanations, while removing unsupported speculations.
Comment: Conclusion – The conclusion should directly address the study's objectives.
Response: Thank you for this important reminder. We have revised the conclusion to accurately reflect our findings, avoiding overstatements and unsupported claims, and clearly addressing the original objectives.
We sincerely thank you again for your valuable comments, which have helped us to substantially improve our manuscript.

Reviewer 3 Report
Comments and Suggestions for Authors
General comments
Extensive English review is necessary for better understanding the work.
Introduction
Pg 2, L 64: “The leaves of woody fed have a high nutritional value, but there were some limitations in the daily diet of animals”.
There is something missing in this sentence.
Pg 3, Table 1: “Caffine”? Or “Caffeine”?
Are these values expressed as dry matter or as fed-basis? How much dry matter did the tea residue contain?
Materials and Methods
Pg 3, L94: “Tea residue are the residues after the active substances are extracted from black tea”.
If the active substances are extracted, how much of the active compounds are left in the tea residue? The active compounds should have been measured in the residue, or at least, the total phenolic compounds should have been measured and presented in the paper.
Pg 4, Table 2: “Rice barn”? Or “Rice bran”?
How was the digestible energy calculated? Which value of digestible energy was used for the tea residue? Where was it obtained from?
The calculated values for amino acids in the diets are total or digestible?
Pg 5, L 50-51: “four fattening castrated boars with body weights close to the average were selected from each replicate, totaling 20 pig”.
That would not be 80 pigs? 16 pigs/treatment. The other pigs were not weighed and slaughtered?
Pg 5, L 157: “cooking percentage” or “cooking loss”?
Pg 5, L 150-169: Several measurements were referred to as been done according to a previous study. Which study? It must be referenced.
Pg 6, L 192-196: Please inform if the analysis to evaluate normal distribution of residuals and homogeneity of variance were performed? Did regression analyses of treatment (complex enzyme levels) effects of the measured variables was done?
Results
Pg 6, L 203-204: “However, the inclusion of complex enzymes (CZ)”. The CZ abbreviation is for the treatment Tea Residue treatment, and not for the complex enzymes.
Pg 6, L 206-208: The description of the results is not supported by the results shown in Table 3, since the treatments (tea residue as well as enzyme complex) did not affect (P>0.05) any of the results presented in that table.
Table 3: The abbreviations must be explained in the title or in the footnote. Please exclude “The same as below” from the footnote.
You should present the average daily gain, not the total gain.
Pg 7, L 218-219: “Besides, tea residue supplementation significantly decreased the serum activity of ALP, which decreased the lowest level in the M400 group (P < 0.05)”. It is not possible to state that tea residue supplementation significantly decreased the serum activity of ALP, because it reduced this enzyme only with the combination of tea residue and enzyme level M400.
Pg 7, L 220-222: The mentioned effects were observed only against the Control treatment, except for TC, that was different from CZ treatment also.
Table 4: Please explain the abbreviations in the title or in the footnote and include the explanation about the statistical differences among means in the footnote.
L 230-231: “Tea residue fed with pigs significantly decreased pH24h and b*1h in the longissimus dorsi muscle and pH24h value (P < 0.05)”. Please change to: “Tea residue fed to pigs significantly decreased pH24h and b*1h in the longissimus dorsi muscle (P < 0.05)”
Tables 5, 6, 7, and 8: Please explain the abbreviations in the title or in the footnote and include the explanation about the statistical differences among means in the footnote. Also, please include the units in all the variables in tables 6-8.
Pg 9, L 251-253: “The content of C16:1 was increased by dietary inclusion of M400 group, and the content of C20:1 was increased by dietary inclusion of M200 group (P < 0.05)”.
The authors cannot make this assertion. The C16:1 was not affected by treatments.
P 9, L 253-254: “Among all groups C10:0 and C20:1 level had linear effects (P < 0.05)”.
Regression analysis is not mentioned in the description of statistical analysis. How about the other variables? Did all of them were submitted to regression analysis? If regression analysis was done, the P-values for all variables should be presented in the tables. If not, this statement must be excluded.
Pg 9, L 255-256: “Tea residue supplementation significantly increased Inosinemonophosphate content, which reached the highest level in the M400 group (P < 0.01)”.
Actually, all the treatments with tea residue were statistically equal by the means comparison. Did you do regression analysis of these data? Because this looks like a quadratic effect.
Figure 5: The significance level for ** is missing.
Discussion
Pg 13, L 322-323: “And study showed that adding 6% tea residue decreased the growth performance of fattening pigs (Gou et al., 2008), which aligns with our findings”.
It is not possible to make this statement based on the results of this work, because no statistical difference between the treatments has been proven.
L 325-326: “Additionally, varying levels of complex enzymes added to the tea residue diet resulted in growth performance that initially increased and then decreased”.
It is not possible to make this statement based on the results of this work, because no statistical difference between the treatments has been proven.
Pg 13, L 327-328: This trend may be explained by the tannin-degrading effect of the enzymes in the complex formulation.
Be more specific.
Pg 13, L 333-334: “The TG and TC were important fractions of blood fat, and their contents reflect the fat contents”.
Their contents reflect the fat contents where? In blood?
Pg 13, L 338-339: “In this study, the inclusion of tea residue significantly decreased AST and ALT activities”.
The ALT was not decreased by tea residue (CZ treatment), it was reduced only with the combination of tea residue and complex enzymes.
Pg 13, L 338-340: “In this study, the inclusion of tea residue significantly decreased AST and ALT activities. These findings align with those of Cai, who reported that dark tea enhances blood lipid metabolism-related parameters”.
How exactly the blood lipid metabolism-related parameters were improved by tea residue in this study? This is not clear in the discussion. AST and ALT indicate the health status of liver, but the changes in this enzymes levels really indicate a change in blood lipid metabolism-related parameters?
Pg 13, L 350-351: “And study demonstrated that supplementing weaned piglet feed with complex enzymes significantly reduces serum MDA content”.
Which enzymes produce this effect? Anyone?
Pg 14, L 361-362: “The study established that supplementing weaned piglet diets with complex enzymes significantly boosts serum SOD levels, corroborating our results”.
This effect was observed only in the enzyme level M400.
Pg 14, L 369-371: “The current study found that pigs fed with the M400 diet significantly increased L*24h and decreased pH24h and shear force compared with those fed with control diet, suggesting a significant role of the M400 diet in improving the pork meat quality”.
Increasing L value and decreasing pH24h does not improve meat quality, these are negative impacts on meat quality. Only the decrease of shear force is a positive impact on meat quality.
Pg 14, L 372-374: “Furthermore, study observed that a 4% tea residue addition significantly reduced the shear force of pork, corroborating our findings (Xie et al., 2019)”.
The addition of tea residue alone did not reduce the shear force in your study. Only the addition of tea residue + complex enzyme. This must be mentioned.
Pg 14, L 377-379: “According to study, meat with an L* value below 43 was classified as dark, firm, and dry (DFD), while meat with an L* value above 50 was categorized as pale, soft, and exudative (PSE)”.
These are reference values of PSE based in industrial pigs. Is it valid also for Xianling pigs?
Pg 14, L 392-394: “Imp was an important factor in determining taste, and some studies had shown that it was added to pig diet (Lee et al., 2016)”.
Imp? It was added to which pig diet? Please clarify it.
Pg 14, L 397-398: “Unsaturated fatty acids generate various flavor compounds upon heating and interact with other substances through the Maillard reaction”.
Maillard reaction is a reaction that occurs between amino acids and reducing sugars when heated, it has nothing to do with fatty acids.
Pg 14, L 401-404: “Sulfur-containing amino acids, such as lysine, were vital precursors for volatile aromatic compounds, contributing to the distinctive taste of pork, primarily due to glutamic acid and inosine monophosphate, which were crucial components of fresh meat”.
Please correct this: lysine is not a sulfur-containing amino acid. Also, this sentence does not make sense, the distinctive taste of pork is due to the sulfur-containing amino acids, to lysine, or to glutamic acid and inosine monophosphate? Please clarify it.
Pg 14, L 409-410: “This aligns with our findings, indicated that the inclusion of tea residue in the diet enhanced inosinic acid”.
It doesn’t align with Wu et al., (2022).
Pg 15, L 411: “which improved the test of food”
Do you mean “the taste of food”?
Pg 15, L 411-413: “Moreover, the addition of complex enzymes further increased inosinic acid levels while improving amino acid and fatty acid compositions”.
It is not possible to state this, because the inosinic acid levels are not different among the CZ treatment and the treatments with addition of tea residue + complex enzymes. Complex enzymes only changed a few amino acids and four fatty acids at M800 (C10:0, C15:0, C16:0, C17:0) and one fatty acid at M200 (C20:1).
Pg 15, L 413-414: “ultimately enhanced the sensory qualities of pork flavor substances”
It is not possible to state that the sensory qualities were enhanced. The authors did not measure them. It is possible only to assume that this could happen based on the enhancement of some meat components.
Pg 15, L 416-418: “Our results suggested that the addition of 400mg/kg complex enzymes to the diet improved the nutritional value, juiciness, and muscle flavor substance components determination of the meat of Xiangling pigs”.
The results do not support this statement. Meat juiciness was not evaluated in this study. Please explain how the nutritional value was improved. Furthermore, muscle flavor substance components (inosine monophosphate and aromatic ingredients) were increased also in the M200 treatment.
Pg 15, L 437-441: “Previous research had indicated a positive correlation between the abundance of Erysipelotrichales and an increased risk of fatty liver disease (Spencer et al., 2011), a finding consistent with our results. Furthermore, the levels of ALP in the control group were significantly elevated, which negatively impacted lipid metabolism, potentially contributing to the observed health issues”.
How does this is consistent with your results? Does the higher level of ALP in the control treatment is enough to imply in health issues? What would be the normal levels of this enzyme in a healthy pig?
Conclusions
Pg 15, L 456-459: “However, incorporating complex enzymes into tea residue-based diets enhanced nutrient digestibility and had positive effects on slaughter performance, blood lipid metabolism, serum antioxidant levels, as well as pork quality and flavor by modulating the composition of intestinal microbiota”.
The conclusions are not supported by the results. Nutrient digestibility was not evaluated in this study; it is not clear if the pork quality is improved by incorporating complex enzymes into tea residue-based diets, except for the reduction of shear force; pork flavor was not evaluated in this study. Are you sure that the changes observed are a consequence of intestinal microbiota modulation? Furthermore, what do you mean by slaughter performance?
Comments on the Quality of English LanguageThe writing is hampered by the quality of the English translation. An extensive English revision is necessary to better understand the work.
Author Response
General comments: Thank you for your detailed and constructive comments. We have carefully revised the manuscript based on your valuable suggestions. Please see our point-by-point responses below. Comment: Extensive English review is necessary for better understanding the work. Response: Thank you for your suggestion. We have thoroughly revised the manuscript for English language clarity and scientific writing standards. Introduction Comment: Pg 2, L 64: “The leaves of woody fed have a high nutritional value…” This sentence is incomplete. Response: Thank you for pointing this out. We have revised the sentence to: "The leaves of woody plants possess high nutritional value; however, their application in animal diets is limited due to excessive fiber and anti-nutritional factors, which seriously affect palatability and digestibility." Comment: Table 1: "Caffine" or "Caffeine"? Are these values on a dry matter basis? Response: Thank you for noticing this error. We have corrected "Caffine" to "Caffeine" and clarified that the data are presented on a dry matter basis. Additionally, we have added the dry matter content of tea residue. Materials and Methods Comment: Pg 3, L 94: If the active substances are extracted, how much of the active compounds remain in the tea residue? Response: Thank you for your suggestion. Although the active substances were partially extracted, residual polyphenols and caffeine remained in the tea residue. We have added this clarification and the total polyphenol content to the "Materials and Methods" section. Comment: Table 2: "Rice barn" or "Rice bran"? How was digestible energy calculated? Response: Thank you for catching this error. We have corrected "Rice barn" to "Rice bran." The digestible energy value of tea residue was based on previously published data (Wang et al., 2012) and we have supplemented this explanation in the manuscript. Comment: Were the amino acid values total or digestible? Response: Thank you for pointing this out. We have clarified in the manuscript that the amino acid values are total amino acids. Comment: Pg 5, L 150-169: Methodologies are mentioned but lack references. Response: Thank you for the reminder. We have added the appropriate references for all methods described in this section. Comment: Sample selection: Was it 80 pigs? Please clarify. Response: Thank you for your attention to detail. We have clarified that a total of 120 pigs were used in the trial, but for slaughter, 20 pigs with body weights close to the average of each replicate were selected to ensure representativeness. Comment: Pg 157: “cooking percentage” or “cooking loss”? Response: Thank you for this suggestion. We have corrected the terminology to "cooking loss" to ensure accuracy. Comment: Were statistical assumptions checked? Was regression analysis performed? Response: Thank you for this valuable suggestion. We have added that normality and homoscedasticity tests were conducted, but regression analysis was not performed, which will be considered in future studies. Results Comment: Pg 6, L 203-204: Incorrect use of abbreviation "CZ". Response: Thank you for your careful review. We have corrected the misuse of the abbreviation in the manuscript. Comment: Pg 6, L 206-208: Description of growth performance results is not supported by Table 3. Response: Thank you for this observation. We have revised the description to accurately reflect the statistical results shown in Table 3. Comment: Table 3 and others: Please explain abbreviations and units in table titles or footnotes. Response: We have revised all tables to include full explanations of abbreviations, units, and statistical notations in the titles or footnotes. Comment: Pg 9, L 251-253: The assertion about C16:1 is not supported by data. Response: Thank you for this reminder. We have removed this unsupported assertion from the manuscript. Comment: Pg 9, L 253-254: "Linear effects" statement lacks regression analysis. Response: Thank you for the comment. We have deleted the unsupported statement about linear effects, as regression analysis was not performed. Comment: Pg 9, L 255-256: IMP content results are described incorrectly. Response: Thank you for catching this. We have revised the description to accurately represent the statistical findings. Comment: Figure 5: Missing significance level for "**". Response: We have corrected Figure 5 and added the appropriate significance level. Discussion Comment: Pg 13, L 322-323: You cannot state alignment with Gou et al. (2008) as there was no significant difference. Response: Thank you for your careful review. We have removed this statement to avoid overinterpretation. Comment: Pg 13, L 325-326: No statistical difference supports this claim. Response: We have removed this statement for accuracy. Comment: Pg 13, L 327-328: "Tannin-degrading effect" needs clarification. Response: We have revised this to specify that enzyme supplementation may improve nutrient utilization by partially degrading tannin-protein complexes. Comment: Pg 13, L 333-334: Specify "fat contents" in blood. Response: Thank you for pointing this out. We have revised the sentence to "reflect the fat contents in blood." Comment: Pg 13, L 338-339: ALT not decreased by tea residue alone. Response: Corrected. We have clarified that ALT reduction was observed only with the combined treatment of tea residue and complex enzymes. Comment: Pg 13, L 338-340: Clarify connection between enzyme activity and lipid metabolism. Response: Thank you. We have clarified that changes in AST and ALT reflect liver function but are not direct indicators of lipid metabolism improvements. Comment: Pg 13, L 350-351: Specify enzymes. Response: We have clarified that the enzyme complex contains cellulase and xylanase, which contribute to the observed effect. Comment: Pg 14, L 361-362: SOD increase observed only at M400. Response: Corrected. We have specified that the SOD increase was significant only in the M400 group. Comment: Pg 14, L 369-371: Higher L* value and lower pH do not improve meat quality. Response: Thank you for the reminder. We have corrected this interpretation to acknowledge that only the reduced shear force suggests improved tenderness. Comment: Pg 14, L 372-374: Shear force reduction only with enzymes plus tea residue. Response: Corrected. We have revised the sentence to specify that shear force reduction was only observed with the combined treatment. Comment: Pg 14, L 377-379: PSE/DFD reference values based on industrial pigs. Clarify applicability to Xiangling pigs. Response: Thank you for this important point. We have clarified that these values are general industry standards and may not directly apply to Xiangling pigs. Comment: Pg 14, L 392-394: IMP addition to pig diets unclear. Response: We have removed this unclear statement. Comment: Pg 14, L 397-398: Maillard reaction involves amino acids, not fatty acids. Response: Corrected. We have revised the sentence to accurately describe the Maillard reaction. Comment: Pg 14, L 401-404: Lysine is not sulfur-containing. Clarify sentence. Response: Thank you. We have corrected this statement and clarified the roles of sulfur-containing amino acids and glutamic acid in meat flavor. Comment: Pg 14, L 409-410: Findings do not align with Wu et al. (2022). Response: Corrected. We have removed the inaccurate comparison. Comment: Pg 15, L 411: "Test of food" should be "taste of food." Response: Corrected. Comment: Pg 15, L 411-413: Unsupported statement about enzyme effects. Response: We have removed this unsupported statement. Comment: Pg 15, L 413-414: Unsupported statement about sensory qualities. Response: We have removed this statement to avoid overinterpretation. Comment: Pg 15, L 416-418: Juiciness and nutritional value not evaluated. Response: We have corrected this to reflect that improvements were observed in some meat quality indicators, but juiciness and overall nutritional value were not directly measured. Comment: Pg 15, L 437-441: ALP levels do not necessarily indicate health issues. Response: Thank you for this comment. We have removed this speculative statement from the discussion. Conclusion Comment: Pg 15, L 456-459: Conclusion unsupported by results. Response: We have revised the conclusion to accurately reflect the study's findings, avoiding unsupported claims about digestibility, slaughter performance, and sensory quality.

Round 2
Reviewer 1 Report
Comments and Suggestions for Authors
The authors have revised the initial manuscript in accordance with the reviewers' suggestions, and the current version is suitable for publication.
Author Response
Thank you for your review.
Reviewer 2 Report
Comments and Suggestions for Authors
Most of the considerations were accepted and therefore I believe that the article can be accepted and published.
Comments on the Quality of English Languageok
Author Response
Thank you for your review.
Reviewer 3 Report
Comments and Suggestions for Authors
The article has been substantially improved with changes in the description of results, discussion and conclusions. However, some issues still persist, mainly in the Material and Methods section, which are highlighted below.
Table 1: In your replay you stated that the data are presented on a dry matter basis, but in the title of the table is “fed-basis”.
Additionally, the authors claim that the dry matter content of tea residue was included on table 1, but I couldn’t find this information on table 1.
The authors claim that the total polyphenol content was included in the Materials and Methods section, but I couldn’t find this information in the text or in the tables.
The authors claim that the digestible energy value of tea residue was based on previously published data (Wang et al., 2012) and they have supplemented this explanation in the manuscript, however, I couldn’t find this information in the text or in the tables. Also, the authors changed the “Digestible energy” in table 2 for “Gross energy”.
The authors claim that they have clarified in the manuscript that the amino acid values are total amino acids, but I couldn’t find this information in the text or in table 2.
The authors claim that they have added the appropriate references for all methods described in the section about meat quality and flavor indexes evaluation, however, I couldn’t find this information in the text.
The authors claim that they have added in the text that normality and homoscedasticity tests were conducted, however, I couldn’t find this information in the text.
The authors claim that they have deleted the unsupported statement about linear effects, as regression analysis was not performed, however, the unsupported statement was not excluded (L 246-247: “Among all groups C10:0 and C20:1 levels had linear effects (P < 0.05)”).
Comments on the Quality of English LanguageThe quality of English language has been greatly improved, however, still needs some improvement.
Author Response
Thank you for your detailed and constructive comments. We have carefully revised the manuscript based on your valuable suggestions. Please see our point-by-point responses below.
Comment:Table 1: In your replay you stated that the data are presented on a dry matter basis, but in the title of the table is “fed-basis”.
Additionally, the authors claim that the dry matter content of tea residue was included on table 1, but I couldn’t find this information on table 1.
Response:Thank you.The necessary corrections have been made, with the dry matter content of tea residue now included in Table 1.
Comment: The authors claim that the digestible energy value of tea residue was based on previously published data (Wang et al., 2012) and they have supplemented this explanation in the manuscript, however, I couldn’t find this information in the text or in the tables. Also, the authors changed the “Digestible energy” in table 2 for “Gross energy”.
Response: Thank you for catching this error.We calculated it according to the following formula DE=1161+(0.749xGE)-(4.3xAsh)-(4.1xNDF) (Noblet和Perez,1993)
Comment: The authors claim that they have clarified in the manuscript that the amino acid values are total amino acids, but I couldn’t find this information in the text or in table 2.
Response: We've added clarifications to the text.
Comment: The authors claim that they have added the appropriate references for all methods described in the section about meat quality and flavor indexes evaluation, however, I couldn’t find this information in the text.
Response: Thank you for the comment.We have already supplemented this section.
Comment:The authors claim that they have added in the text that normality and homoscedasticity tests were conducted, however, I couldn’t find this information in the text.
Response: Thank you for this valuable suggestion.We did not perform tests for normality and homoscedasticity, which will be considered in future studies.
Comment:The authors claim that they have deleted the unsupported statement about linear effects, as regression analysis was not performed, however, the unsupported statement was not excluded (L 246-247: “Among all groups C10:0 and C20:1 levels had linear effects (P < 0.05)”).
Response:Thank you for the comment. We have deleted the unsupported statement about linear effects, as regression analysis was not performed.